# Cerebello-Thalamo-Cortical MR Spectroscopy in Patients with Essential Tremor Undergoing MRgFUS Thalamotomy: A Pilot Study

**DOI:** 10.3390/life12111741

**Published:** 2022-10-29

**Authors:** Federico Bruno, Emanuele Tommasino, Alessia Catalucci, Veronica Piccin, Antonio Innocenzi, Maria Ester Carugno, Filippo Colarieti, Leonardo Pertici, Antonio Di Gioia, Claudia D’Alessandro, Cristina Fagotti, Patrizia Sucapane, Francesca Pistoia, Pierpaolo Palumbo, Francesco Arrigoni, Ernesto Di Cesare, Carmine Marini, Antonio Barile, Alessandra Splendiani, Carlo Masciocchi

**Affiliations:** 1Emergency Radiology, San Salvatore Hospital, 67100 L’Aquila, Italy; 2Italian Society of Medical and Intervention Radiology (SIRM), SIRM Foundation, 20122 Milan, Italy; 3Department of Biotechnological and Applied Clinical Sciences, University of L’Aquila, 67100 L’Aquila, Italy; 4Neuroradiology and Interventional Radiology, San Salvatore Hospital, 67100 L’Aquila, Italy; 5Neurology, San Salvatore Hospital, 67100 L’Aquila, Italy; 6Department of Diagnostic Imaging, Area of Cardiovascular and Interventional Imaging, Abruzzo Health Unit 1, 67100 L’Aquila, Italy; 7Department of Life, Health and Environmental Sciences, University of L’Aquila, 67100 L’Aquila, Italy

**Keywords:** MR spectroscopy, tremor, Parkinson’s disease, MRgFUS, thalamotomy

## Abstract

Previous literature studies explored the association between brain neurometabolic changes detected by MR spectroscopy and symptoms in patients with tremor, as well as the outcome after deep brain stimulation (DBS) treatment. The purpose of our study was to evaluate the possible changes in cerebello-thalamo-cortical neurometabolic findings using MR spectroscopy in patients submitted to MRgFUS thalamotomy. For this pilot study, we enrolled 10 ET patients eligible for MRgFUS thalamotomy. All patients were preoperatively submitted to 3T MR spectroscopy. Single-voxel MRS measurements were performed at the level of the thalamus contralateral to the treated side and the ipsilateral cerebellar dentate nucleus. Multivoxel acquisition was used for MRS at the level of the contralateral motor cortex. At the 6-month follow-up after treatment, we found a statistically significant increase in the Cho/Cr ratio at the level of the thalamus, a significant increase of the NAA/Cr ratio at the level of the dentate nucleus and a significant decrease of the NAA/Cho ratio at the level of the motor cortex. We found a significant positive correlation between cortical NAA/Cr and clinical improvement (i.e., tremor reduction) after treatment. A significant negative correlation was found between clinical improvement and thalamic and cerebellar NAA/Cr. Confirming some previous literature observations, our preliminary results revealed neurometabolic changes and suggest a possible prognostic role of the MRS assessment in patients with ET treated by MRgFUS.

## 1. Introduction

In the recent years, several trials have proved the safety and efficacy of magnetic-resonance-guided focused ultrasound (MRgFUS) thalamotomy for the treatment of tremor in Parkinson’s disease (PD) and essential tremor (ET), mainly with ventral intermediate (Vim) ablation [1,2,3,4,5,6,7]. Although functional MRI has limited application in the procedural phases of MRgFUS thalamotomy, advanced MRI sequences have shown promising results for the screening of patients fit for undergoing the treatment and for lesion evaluation after the procedure. Several papers have tried to explore the possible role of advanced sequences such as DWI, DTI and fMRI for the pretreatment and outcome evaluation [8,9,10,11,12,13]. For instance, DWI provides quantitative information about the soft tissue microstructure based on differences in water proton mobility and cell density assessment, and it has been used for evaluating the effectiveness of thalamotomy by assessing the ADC values of the thalamotomy lesion and related white matter tracts. Similarly, the DTI and the FA values have been used for the direct and indirect targeting of the thalamus nuclei, giving a detailed presentation of the main white matter fiber tracts. Finally, blood-oxygenation-dependent imaging (BOLD), which is a technique used to produce functional MRI (fMRI) images that are the result of changes in regional blood concentrations of oxy- and deoxy-hemoglobin, has been used for the evaluation of the activation of different brain regions in PD and ET [2,14,15,16,17]. Notwithstanding the interest in these novel techniques, no paper has dealt with the possible role of magnetic resonance spectroscopy (MRS) in MRgFUS thalamotomy.

MRS is a specialized MRI technique that has been used for probing the metabolism of well-defined regions of the human body, obtaining in vivo and in situ concentration measures for certain chemicals in complex samples [18].

In the CNS, it allows the identification of a number of cerebral metabolites such as the N-acetyl-aspartate (NAA) which is found in high concentrations in neurons and is a marker of neuronal viability, creatine (Cr), an important molecule for the storage and transfer of energy in metabolically active tissues and choline compounds (Cho), which are important metabolites for the cellular membrane turnover.

The NAA/Cr ratio is considered to be a metabolic marker for neuronal function, and a reduction of this ratio indicates a damage or degeneration of neuronal and/or axonal structures [19].

A few studies aimed to assess the prognostic value of the concentration of certain brain metabolites after deep brain stimulation of the subthalamic nucleus (STN DBS) or after dopaminergic medical treatment [20,21]. They especially examined the cortical function, without probing the thalamus and other important nuclei. In addition, no studies have been performed before and after MRgFUS thalamotomy to evaluate the metabolic changes in the motor cortex, in the thalamus and other brain regions.

Therefore, the aim of our study was to investigate the metabolic and neurochemical changes in the motor cortex (MC), in the thalamus (T) and in the dentate-rubro-thalamo-cortical tract (DRTT) using MRS in patients with ET submitted to MRgFUS Vim thalamotomy.

## 2. Materials and Methods

### 2.1. Study Population

We evaluated 10 consecutive ET patients (10 males, mean age 70.21 ± 7.5 years, range 50–81) eligible for unilateral MRgFUS Vim ablation at the ‘‘San Salvatore” Hospital in L’Aquila. Patients were evaluated for inclusion criteria by two expert neurologists (PS and FP) of the Movement Disorders Clinic based at the same Institution. All of them were right-handed and a left thalamotomy was the indicated treatment. A pretreatment clinical examination included the collection of demographic data, a detailed clinical history, and the assessment of tremor severity by the Fahn–Tolosa–Marin (FTM) tremor rating scale. The FTM test was also administered the day after the treatment, and after 6 months.

Inclusion and exclusion criteria were the same as the ones for the treatment selection and they are described elsewhere. Written informed consent was obtained from all subjects who met the eligibility for the treatment. The present study was approved by the IRB of the University of L’Aquila (protocol number: 01/2020).

### 2.2. MRI Examination and Procedural Parameters

All examinations were performed using a 3-tesla MR scanner (MR750w, GE Healthcare, Chicago, IL, USA) with a 32-channel head coil 1 month before and 6 months after the treatment. The protocol included FLAIR (slice 3–0.3, TR 11,000, frequency FoV 24, phase FoV 0.8), GRE (slice 3–0.3, TR 960, frequency FoV 26, phase FoV 0.75), SWI (slice 2 mm, frequency FoV 24, phase FoV 0.85), DWI (slice 3–0.3, TR 10,550, frequency FoV 26, phase FoV 0.8) sequences on axial planes, T2-weighted (slice 3.0–0.3, TR 7854, frequency FoV 26, phase FoV 0.8) sequences on coronal and axial planes, and a volumetric T1-weighted 3D-IR-FSPGR (BRAVO) sequence (slice 1 mm, TR 8.5, frequency FoV 25.6, phase FoV 0.8). The MRS protocol included:


two single-voxel (SV) sequences (TR 1700, TE 30, TM, voxel size 15 × 15 × 15, water suppression Hz, acquisition duration 4 m) with the 1st ROI placed to the contralateral T and the 2nd in the homolateral DN to the treated side.one multivoxel (SE) sequence (TR 1700, TE 30, TM, including the primary motor cortex 120 × 120, water suppression Hz, acquisition duration 6 m) with 2 ROIs placed on the hand knob area in the primary motor cortex bilaterally (Figure 1).


The NAA/Cr, the Cho/Cr, the NAA/Cho and the GABA/Cr ratios were obtained on a dedicated workstation. These ratios were correlated to the FTM score after the treatment and the procedural parameters.

The latter ones were acquired during the sonications and they were described in our previous work [15]. In particular, we took into consideration the number of sonications (nos) and the number of movements (nom) taken for an effective treatment. The sonication procedure involved three steps. The first (alignment) included short sonications with very low energy (1500–3000 J), so that the temperature increase was visible in the thermometric maps without creating biological effects. At this stage, temperatures of 40–45 °C were reached (the thermometric maps were acquired every 3 s and provided two values: the average temperature and the maximum temperature for a spot of about nine voxels). The second step (verification) included sonications with increasing energy and power parameters to reach higher temperatures (46–52 °C) to obtain a neuromodulation effect, confirming the efficacy of the treatment in the target and the possible presence of adverse effects. The clinical response was controlled both during sonications to assess the reduction/disappearance of the tremor and at the end of each sonication to assess the appearance of adverse effects. The “nos” included all the sonications performed in these three steps.

In addition, based on the clinical response, it was possible to move the target, taking into account the somatotopic distribution of the neurons of the Vim and the neighboring structures which gave the “nom”.

### 2.3. Statistical Analysis

Data analyses were performed using MedCalc (Version 20.111). Qualitative variables were summarized as frequency and proportions. The values of continuous variables were tested for normal distribution using Shapiro–Wilk’s test and reported as means and standard deviations (SD) or medians and interquartile ranges (IQRs) according to their distribution. The differences of quantitative values (age, FTM, MoCA, Cho/Cr ratio and the NAA/CR ratio) between groups were compared using the Wilcoxon test or the Student t-test according to their distribution. The variance of FTM scores at the different follow-up times was evaluated using an ANOVA for repeated measures. The point biserial correlation was applied to evaluate the correlation between continuous and binomial variables. A correlation analysis of continuous variables was performed by a Spearman correlation test.

## 3. Results

### 3.1. Thalamus

Before the treatment, at the level of the thalamus, the median NAA/Cr ratio was 1.69 (max. 1.81 and min. 1.43, CI 95% 1.51–1.76), the Cho/Cr ratio was 0.91 (max. 1.160 and min. 0.810, CI 95% 0.83–1.024) and the GABA/Cr ratio was 0.64 (max. 0.93 and min. 0.46, CI 95% 0.52–0.82). At the 6-month follow-up, at the level of the thalamus, the median NAA/Cr ratio was 1.90 (max. 2.80 and min. 1.46, CI 95% 1.49–1.90), the Cho/Cr ratio was 1.14 (max. 1.40 and min. 0.99, CI 95% 1.03–1.36) and the GABA/Cr ratio was 0.56 (max. 0.66 and min. 0.40, CI 95% 0.44–0.65). See the details in Table 1.

### 3.2. Cerebellum (Dentate Nucleus)

Before the treatment, at the level of the cerebellum, in the DN, the median NAA/Cr ratio was 1.53 (max. 1.79 and min. 1.36, CI 95% 1.23–2.07), the Cho/Cr ratio was 0.83 (max. 1.01 and min. 0.83, CI 95% 0.80–1.05) and the GABA/Cr ratio was 0.64 (max. 0.80 and min. 0.53, CI 95% 0.57–0.72). After the treatment, at the level of the cerebellum, in the DN, the median NAA/Cr ratio was 2.45 (max. 2.62 and min. 2.15, CI 95% 1.23–2.07), the Cho/Cr ratio was 1.25 (max. 1.44 and min. 1.16, CI 95% 0.80–1.05). See details in Table 2.

### 3.3. Motor Cortex (Hand Knob)

Before the treatment, at the level of the hand knob in the left motor cortex, the median NAA/Cr ratio was 2.96 (max. 17.33 and min. 2.00, CI 95% 0.87–7.52), the NAA/Cho ratio was 3.44 (max. 9.11 and min. 2.11, CI 95% 2.46–5.47) and the Cho/Cr ratio was 2.96 (max. 3.35 and min. 2.05, CI 95% 2.42–3.24). After the treatment, at the level of the hand knob in the motor cortex, the median NAA/Cr ratio was 2.28 (max. 3.77 and min. 1.77, CI 95% 1.84–2.77), the NAA/Cho ratio was 2.92 (max. 4.63 and min. 1.21, CI 95% 1.69–3.17) and the Cho/Cr ratio was 2.12 (max. 2.99 and min. 1.70, CI 95% 1.83–2.63). See details in Table 3.

Similarly, before the treatment, at the level of the hand knob in the right motor cortex, the median NAA/Cho ratio was 2.02 (max. 13.90 and min. 0.73, CI 95% 1.73–2.81), the NAA/Cr ratio was 1.91 (max. 2.58 and min. 0.04, CI 95% 1.46–2.47) and the Cho/Cr ratio was 0.86 (max. 1.91 and min. 0.68, CI 95% 0.69–1.54). After the treatment, at the level of the hand knob in the right motor cortex, the median NAA/Cr ratio was 2.22 (max. 4.30 and min. 1.07, CI 95% 1.17–2.27) (*p* = 0.76), the NAA/Cho ratio was 2.25 (max. 6.91 and min. 1.77, CI 95% 1.86–3.13) (*p* = 0.46) and the Cho/Cr ratio was 0.766 (max. 1.11 and min. 0.35, CI 95% 0.43–0.95) (*p* = 0.10).

The analysis of the difference in the variations of Cho and NAA between the two hemispheres did not show any statistical significance. In fact, comparing the Cho in the right cortex with the same parameter in the left cortex through the Mann–Whitney test, we obtained a *p*-value of 0.6857. The behavior of the NAA was similar (*p* = 0.48).

### 3.4. Correlation and Simple Regression Analysis

A correlation analysis was performed between the MRS metabolic scores before and after treatment, the ∆(Pre-Post) FTM rating scale and the procedural parameters. Two separate analyses were carried out, the first involving the premetabolic ratios in order to evaluate their effect on the clinical outcome, the second involving the ∆ between the pre- and postmetabolic ratios in order to evaluate their modification after the MRgFUS thalamotomy.

The mean pretreatment FTM value was 39.75 (CI 95% 25.87–53.63) while the mean post-treatment FTM value was 18.125 (CI 95% 9.67–27.94) (*p* < 0.05).

First, the correlation was calculated to evaluate the relationship between the ∆(Pre-Post) FTM values and the ∆(Pre-Post) NAA in the thalamus, in the DN (DRTT) and in the motor cortex (hand knob).

A significant positive correlation was found (R = 0.5, *p* < 0.04) between the motor cortex ∆(Pre-Post) NAA and the ∆(Pre-Post) FTM. It was found that higher NAA values in the cortex were correlated to higher ∆(Pre-Post) FTM.

On the other hand, no significant correlation was found (R = 0.7, *p* < 0.49) between the thalamic NAA/Cr ratio as well as (R = 0.31, *p* < 0.18) between the DN NAA/Cr ratio and the ∆(Pre-Post) FTM.

Secondly, another significant positive correlation was found (R = 0.5, *p* < 0.03) between the motor cortex ∆(Pre-Post) Cho and the ∆(Pre-Post) FTM.

Finally, a correlation was also calculated to evaluate the relationship between the ∆(Pre-Post) FTM and the pre-GABA/Cr in the thalamus and in the DN. No significant relationship was found in this case.

We also tried to identify a correlation between the premetabolic ratios and the procedural parameters, particularly with the number of sonications (nos) and the number of movements (nom).

The mean number of sonications was 10.42 (CI 95% 9.3–11.4, range 9–12).

The mean number of movements from the target was 10.42 (CI 95% 9.3–11.4, min. one and max. three).

There was a significant negative correlation (r = 0.7, *p* < 0.05) between the GABA/Cr and the nom in the thalamus.

A correlation analysis between the brain volumes at baseline (before the treatment) with age, tremor and cognitive function did not show any significant results in both PD and ET. Similarly, the point biserial correlation, used to assess whether or not volume variation was linked to a worse clinical outcome (i.e., tremor relapse), did not show any significant result.

## 4. Discussion

Nowadays, magnetic-resonance-guided focused ultrasound (MRgFUS) thalamotomy for the treatment of tremor in Parkinson’s disease (PD) and essential tremor (ET) is considered, along with DBS, the most effective therapeutic approach for these kinds of conditions if the medical management has failed.

Several factors that may contribute to the clinical outcome of MRgFUS have been extensively studied, namely the skull density ratio (SDR) and brain volume [2,15].

Nevertheless, others are still under consideration. For this reason, to our knowledge, this is the first study applying MRS to the pre- and post-treatment evaluation of ET patients with clinical improvement following MRgFUS thalamotomy.

Interestingly, in our preliminary study, we found that cortical ∆(Pre-Post) NAA was positively correlated with the clinical outcome (FTM). Particularly, a higher pre-NAA corresponded to higher ∆FTM values 24 h after the MRgFUS thalamotomy.

This may be explained by the role of the NAA as a marker of neuron viability. In fact, Louis ED et al., in 2002, found that lower values of the NAA/Cr ratio in the cerebellar cortex were accompanied by a more severe tremor [22]. Therefore, a lower NAA ratio may reflect a compromised cerebral area, with reduced neuronal density, and as a consequence, worse clinical scores and worse prognosis. That is why in our study, higher NAA values were accompanied by higher ∆FTM values since it reflected a better neuronal viability in the motor cortex. This may also explain the positive relationship between the ∆(Pre-Post) choline at the cortical level and the ∆(Pre-Post) FTM.

In addition, we explored the relationship between the MRS ratios and the procedural parameters, particularly the number of sonications and the number of movements from the target in order to obtain an effective ablation.

We found an inverse relationship between the GABA/Cr ratio in the thalamus and the number of movements from the target to obtain an effective response. A lower GABA/Cr ratio corresponds to a higher number of movements. This may be due to the fact that sonications at non-ablative temperatures determine different effects of neuromodulation on metabolically damaged neurons (low GABA/Cr ratio). In normal conditions, neuromodulation has excitatory effects on GABAergic neurons. Therefore, in the thalamus where the concentration of these neurons is reduced, the neuromodulation response could be lower and the MRgFUS operator will have to change the sonications’ focus. Finally, these results are in line with other research papers that found lower values of the GABA/Cr ratio in the thalamus and the motor cortex of patients with PD and ET and that this was accompanied by worse clinical scores [23].

This study is still in progress and the number of patients is too scarce for any definitive prognostic role of these values. In addition, we studied the correlation with the post-FTM at 24 h and 6 months after the treatment and a further evaluation is needed for assessing the metabolic changes after this time frame. For example, in 2008, Llumiguano et al. demonstrated using MRS that the cortical (LFBC) NAA/Cho and NAA/Cr ratios were significantly increased, paired with a significant decrease of the Cho/Cr ratio in all patients treated with STN DBS [21], while Lucetti et al. demonstrated the restoration of the Cho/Cr ratio in the motor cortex of PD patients treated with the dopamine agonist pergolide after 7 months [20]. We can still hypothesize explanations for our results. The increase in metabolic indices at the level of the treated thalamus is probably due to the repair processes following the thalamotomy lesion. On the other hand, the finding of the reduction of metabolic indices at the cortical level was unexpected. However, it must be considered that there is evidence in the literature according to which the GABAergic deficit due to Purkinje cell degeneration may lead to a reduction of the inhibitory output from the cerebellar deep nuclei to the thalamus, with a consequent increase in the excitatory output to the motor cortex. This also results in increased cortical output from the motor cortex. The reduction in tremor after thalamotomy treatment could therefore be the cause of this reduction in cortical hyperactivity [24].

## 5. Conclusions

In recent years, several trials have proved the utility of advanced MRI techniques, such as DWI, DTI and fMRI, for the screening and outcome evaluation of magnetic-resonance-guided focused ultrasound (MRgFUS) thalamotomy for the treatment of tremor in Parkinson’s disease (PD) and essential tremor (ET), mainly with ventral intermediate (Vim) ablation. Nonetheless, no evidence has been provided for the utility of MRS, and no studies on the brain metabolic changes after the treatment have been conducted. We proved that neurometabolic changes appeared after MRgFUS thalamotomy, particularly in the dentate nucleus and in the motor cortex, and that these changes were related to the treatment itself. This may also suggest a possible prognostic role of the MRS assessment in patients with ET treated by MRgFUS. Nonetheless, two important aspects need to be taken into consideration. First, MRS is not a widely available technique. Secondly, a bigger cohort is necessary to delineate clear spectroscopic cut-off results in order to predict the actual prognostic validity.

## Figures and Tables

**Figure 1 life-12-01741-f001:**
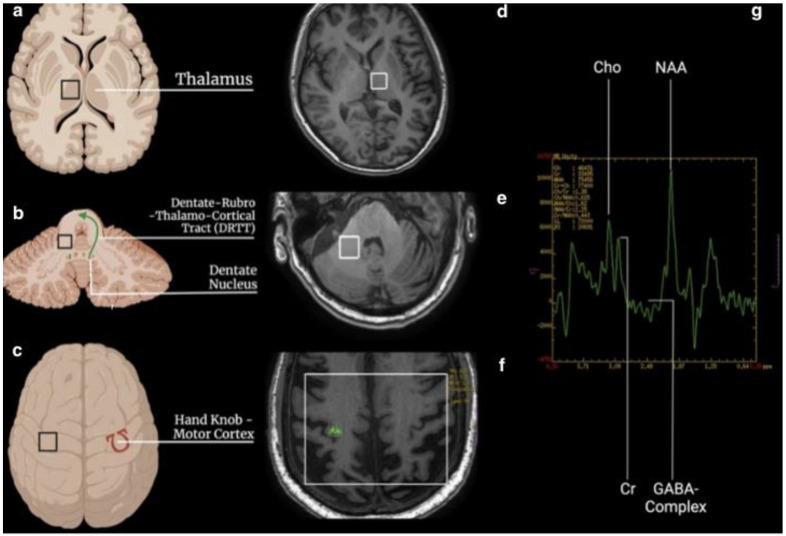
Schematic representation of ROI placement for MRS acquisition. (**a**–**c**): localization of the regions of interest (black square) in the thalamus, dentate nucleus, and hand knob, respectively. The green arrow represents the DRTT, while the omega sign Ω in red is the hand knob. (**d**–**f**): the same localization of the region of interest (white square) on MRI brain images. (**g**): spectroscopy curve with the peak of choline (Cho), creatine (Cr), N-acetyl-aspartate (NAA), and GABA-complex.

**Table 1 life-12-01741-t001:** The Cho/Cr increase was significant (*p* = 0.0314).

	Thalamus					
	NAA/Cr Ratio		Cho/Cr Ratio		GABA/Cr Ratio	
	Pretreatment	6 M Follow-up	Pretreatment	6 M Follow-up	Pretreatment	6 M Follow-up
**Min.**	1.43	1.46	0.81	0.99 *	0.58	0.40
**Max.**	1.81	2.80	1.16	1.40	0.93	0.66
**Median**	1.69	1.90	0.91	1.14	0.66	0.54
**CI 95%**	1.51–1.76	1.49–1.90	0.83–1.02	1.03–1.36 *	0.45–0.96	0.35–0.74

* *p*-value < 0.05.

**Table 2 life-12-01741-t002:** The Cho/Cr increase was significant (*p* = 0.0271).

	Dentate Nucleus (Cerebellum)			
	NAA/Cr Ratio		Cho/Cr Ratio	
	Pretreatment	6 M Follow-up	Pretreatment	6 M Follow-up
**Min.**	1.12	1.30 *	0.69	0.80
**Max.**	2.11	2.62 *	1.21	1.44
**Median**	1.53	2.15 *	1.01	1.16

* *p*-value < 0.05.

**Table 3 life-12-01741-t003:** The NAA/Cho decrease was significant (*p* = 0.0271).

	Motor Cortex					
	NAA/Cr Ratio		Cho/Cr Ratio		NAA/Cho Ratio	
	Pretreatment	6 M Follow-up	Pretreatment	6 M Follow-up	Pretreatment	6 M Follow-up
**Min.**	2.00	1.70	2.050	1.70	2.11	1.21
**Max.**	17.33	3.77	3.35	2.99	9.11	4.63
**Median**	2.96	2.28	2.96	2.12	3.44	2.92 *
**CI 95%**	0.87–7.52	1.88–2.77	2.42–3.24	1.83–2.63	2.46–5.47	1.69–3.17

* *p*-value < 0.05.

## Data Availability

The datasets generated during and/or analysed during the current study are available from the corresponding author on reasonable request.

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
