# Peer review of "Cerebello-Thalamo-Cortical MR Spectroscopy in Patients with Essential Tremor Undergoing MRgFUS Thalamotomy: A Pilot Study"

_life, 2022, doi:10.3390/life12111741_

Round 1
Reviewer 1 Report
This is a pilot study applying MR Spectroscopy in patients with essential tremor undergoing MRgFUS thalamotomy. Clinical improvement following the MRgFUS Thalamotomy was correlated with metabolic MRS ratios pre and post treatment, and some correlations were found for some of the investigated areas. This is a clearly written paper with solid method, while the authors explicitly state that this is a pilot study with a small number of patients and avoid to overtranslate their results.
My main concern is about the whole concept of using MRS as a predictive tool for patient selection for MRgFUS Thalamotomy especially considering that this is not a widely available technique. I guess for this to enter the clinical arena we should have clear cut results in a lagre series of patients with specif cut offs predicting the bad responders and much longer follow-ups. I think this should be somehow stated in the conclusions.
Author Response
We are grateful to the reviewer for their insightful comments on my paper. We have been able to incorporate changes to reflect most of the suggestions provided by the reviewers. We have highlighted the changes within the manuscript .
Here is a point-by-point response to the reviewers’ comments and concerns.
My main concern is about the whole concept of using MRS as a predictive tool for patient selection for MRgFUS Thalamotomy especially considering that this is not a widely available technique. I guess for this to enter the clinical arena we should have clear cut results in a lagre series of patients with specif cut offs predicting the bad responders and much longer follow-ups. I think this should be somehow stated in the conclusions.
RESPONSE:
Thank you for pointing this out. We agree with this comment. Therefore, we stated that in the conclusions (highlighted in yellow)
Reviewer 2 Report
1. Most items in the three tables are similar, so it is better to merge three tables into one.
2. The figure is very difficult to understand. The graph is composed of three parts with abbreviation (Cho, NAA…) and Icons (small white boxes, small green dots, small red circle etc.) which should be described clearly in the figure legend. The authors should mark A, B or C on the top and mark left or right on the two sides of the figure respectively.
Author Response
We are grateful to the reviewers for their insightful comments on my paper. We have been able to incorporate changes to reflect most of the suggestions provided by the reviewers. We have highlighted the changes within the manuscript.
Here is a point-by-point response to the reviewers’ comments and concerns.
Most items in the three tables are similar, so it is better to merge three tables into one.Response: Thank you for this suggestion. Howver, we would keep this layout because it looks more precise for us. Thank you for your understanding.
The figure is very difficult to understand. The graph is composed of three parts with abbreviation (Cho, NAA…) and Icons (small white boxes, small green dots, small red circle etc.) which should be described clearly in the figure legend. The authors should mark A, B or C on the top and mark left or right on the two sides of the figure respectively.
Response: We agree with this and have incorporated your suggestion in the figure legend.
